# Rising Atmospheric Temperature Impact on Wheat and Thermotolerance Strategies

**DOI:** 10.3390/plants10010043

**Published:** 2020-12-27

**Authors:** Adeel Khan, Munir Ahmad, Mukhtar Ahmed, M. Iftikhar Hussain

**Affiliations:** 1Department of Plant Breeding and Genetics, PMAS-Arid Agriculture University Rawalpindi, Rawalpindi 46300, Pakistan; adeelkhanbreeder@gmail.com (A.K.); muneer.ahmad@uaar.edu.pk (M.A.); 2Department of Agricultural Research for Northern Sweden, Swedish University of Agricultural Sciences, 90183 Umeå, Sweden; 3Department of Agronomy, PMAS Arid Agriculture University, Rawalpindi 46300, Pakistan; 4Department of Plant Biology & Soil Science, Faculty of Biology, University of Vigo, Campus As Lagoas Marcosende, 36310 Vigo, Spain; mih786@gmail.com; 5CITACA, Agri-Food Research and Transfer Cluster, Campus da Auga, University of Vigo, 32004 Ourense, Spain

**Keywords:** heat stress, photosynthesis, antioxidant enzymes, HSPs, QTLs, omics

## Abstract

Temperature across the globe is increasing continuously at the rate of 0.15–0.17 °C per decade since the industrial revolution. It is influencing agricultural crop productivity. Therefore, thermotolerance strategies are needed to have sustainability in crop yield under higher temperature. However, improving thermotolerance in the crop is a challenging task for crop scientists. Therefore, this review work was conducted with the aim of providing information on the wheat response in three research areas, i.e., physiology, breeding, and advances in genetics, which could assist the researchers in improving thermotolerance. The optimum temperature for wheat growth at the heading, anthesis, and grain filling duration is 16 ± 2.3 °C, 23 ± 1.75 °C, and 26 ± 1.53 °C, respectively. The high temperature adversely influences the crop phenology, growth, and development. The pre-anthesis high temperature retards the pollen viability, seed formation, and embryo development. The post-anthesis high temperature declines the starch granules accumulation, stem reserve carbohydrates, and translocation of photosynthates into grains. A high temperature above 40 °C inhibits the photosynthesis by damaging the photosystem-II, electron transport chain, and photosystem-I. Our review work highlighted that genotypes which can maintain a higher accumulation of proline, glycine betaine, expression of heat shock proteins, stay green and antioxidant enzymes activity viz., catalase, peroxidase, super oxide dismutase, and glutathione reductase can tolerate high temperature efficiently through sustaining cellular physiology. Similarly, the pre-anthesis acclimation with heat treatment, inorganic fertilizer such as nitrogen, potassium nitrate and potassium chloride, mulches with rice husk, early sowing, presoaking of a 6.6 mM solution of thiourea, foliar application of 50 ppm dithiothreitol, 10 mg per kg of silicon at heading and zinc ameliorate the crop against the high temperature. Finally, it has been suggested that modern genomics and omics techniques should be used to develop thermotolerance in wheat.

## 1. Introduction

Climate change is the result of a higher level of greenhouse gases such carbon dioxide (CO_2_), nitrous oxide, and methane (CH_4_). These gases can entrap the sun rays leading towards the severity of extreme events for crops development [1,2]. It has been observed that CO_2_ was increased 0.6 ± 0.1 ppm/year in the early 1960s and 2.3 ± 0.6 ppm/year during the last decade. Meanwhile, the CH_4_ gas was doubled after the industrial revolution until the 1980s and it is increasing at the rate of 12 parts per billion per year. However, during the last three decades it was increasing 2–5 parts per billion per year. The nitrous oxide concentration was enhanced 18% more than the 1970s and increased 0.8 parts per billion per year [3,4].

The escalating global warming evokes an extreme weather pattern, increases disease incidences, insect pest survival, and ultimately influences crop productivity [5,6]. Global warming potential (GWP) is the contribution of one molecule of compound over 100 years to global warming as compared to CO_2_. It allows the comparison of different gas contributions to global warming and how much energy emissions of 1 ton of gas absorbs more than 1 ton of CO_2_ over a given time period. The larger global warming potential represents more potential of the given gas to persist and the ability to warm the Earth temperature over a given time period. The GWP of carbon dioxide is 1, CH_4_ 28–36, and nitrous oxide 265–298 over 100 years. However, these gases possess more potential and persistency to entrap the sun rays than CO_2_ but a major contributor in global warming is CO_2_ [7].

Agricultural crop productivity depends on biotic (diseases and insect pest) and abiotic (heat, drought, and salinity) factors [8]. Among the abiotic stresses, the higher temperature is a major concern influencing crop growth and development. The global temperature roughly increased by 1.5 °C with the same accelerating trend in all regions from the 1970s, as reported by the intergovernmental panel of climatic change (IPCC) and was predicted to increase 2.5–5.8 °C until the 2100s [3]. The global average temperature annually increased by 0.04–0.07 °C and 0.15–0.17 °C per decade since the 1880s and 1970s, respectively according to the National Oceanic and Atmospheric Administration (NOAA, 2018). Therefore, global warming characterized by an extreme temperature possesses the challenge to improve the yield potential of crops.

Terminal and continual heat stresses are two major constraints influencing crops growth and development. The temperature threshold levels were reported at different stages for crops viz., cotton [9,10,11,12,13], rice [14,15,16,17,18,19], sorghum [20,21], barley [12], maize [9,22,23], and soybean [23]. Wheat is an imperative staple food, the cheapest energy source, provides 8–20% of protein, and 70–75% of calories in our average diet [24], but a high temperature restricts the wheat crop to express its full genetic potential. Therefore, there is a dire need to understand the wheat response against the high temperature and a suitable strategy to improve its productivity. 

## 2. Impact of High Temperature on Wheat

High temperature influences the wheat productivity in tropical, subtropical, arid and semi-arid regions of the world. The optimum temperature for wheat growth and development are given in Table 1. The high temperature in the tropical region is an inevitable constraint for wheat during germination and early growth stages, whereas in the Mediterranean region, the reproductive stage is highly sensitive [25]. A high temperature of 3–4 °C above the optimum temperature at grain filling reduces 10–50% of the wheat yield in Asia with the current production technology and varieties [26]. High temperature declines 0.07% per °C grain yield depending on the wheat variety [27]. Each degree increase in the temperature at the grain filling duration reduces 6% of wheat yield globally [28,29] and 3–17% in South Asia including India and Pakistan [30]. It accredited directly or indirectly the disturbance in different cellular, physiological functions and metabolic pathways associated with the grain yield in wheat (Figure 1).

### 2.1. Cellular Metabolism

The plasma membrane is a highly organized structure composed of lipids and proteins. It regulates the enzymatic activity and transport of ions. High temperature alters the microtubules organization, expansion, elongation, and cell differentiation [48]. It increases the kinetic energy of hydrogen bonds between adjacent fatty acids, weakens the bonds, and leads to membrane fluidity. This fluidity, unsaturation of fatty acids, and disruption of different proteins trigger the electrolyte leakage [49,50]. High temperature causes 25–55% electrolyte leakage at 45 °C for 1 h [51], while 21–40% leakage at 40 °C for 30 min [52]. Therefore, the cell damages its internal composition and sustainable physiological processes (e.g., photosynthesis, respiration, and transpiration) associated with the synthesis and translocation of carbohydrates into the grains.

### 2.2. Grain Filling Duration

High temperature enforces the plant to complete the growing degree days earlier, which results in early maturity and shorter life cycle of plant, lesser biosynthetic products accumulation, and ultimately poor grain development [32,53,54]. Vernalization (*VRN1, VRN2*) and the photoperiodic (*PPD-A1, PPD-D1*) sensitive gene determines the developmental phases at volatile temperature events and triggers earliness in wheat by limiting various growth phases [55,56]. The longer grain filling duration determines the appropriate grains development associated with the grain yield [57]. However, high temperature reduces the duration to uptake the available nutrients and translocation of photosynthates.

### 2.3. Grain Formation and Development

Vital events at the reproductive stage such as flowering initiation, pollen germination, pistil receptiveness, and embryo development determine the florets fertility [58,59]. The embryo sac and embryo formation are sensitive to high temperature [60]. Microgametogenesis and microsporogenesis are sensitive to high temperature, which hinder the gametes development and cause spores abortion [61,62]. 

Wheat grain contains 60–70% starch content and gradually drops under high temperature [63,64]. High temperature inhibits the starch accumulation into grains ascribable to the enzymes inactivity viz., granule bound starch, soluble starch, and sucrose synthase during the grain filling phase [65,66]. It also declines the starch content synthesis [67,68], stem reserves carbohydrates translocation [69,70], alters the structure of aleurone layer, and endosperm of seed [71,72], which ultimately influences grain development.

### 2.4. Leaf Senescence

Leaf senescence is the reduction in green leaf area during the reproductive phase due to the retardation in the chlorophyll content and carotenoids [73,74]. The chlorophyll content and carotenoid have an indispensable role in harvesting sunlight for photosynthesis [75]. High temperature disturbs the chloroplast integrity, leaf senescence, and ultimately photosynthesis in wheat [76]. 

Leaf senescence during the grain filling duration degrades the leaf chlorophyll content. Initially, chlorophyll-b is converted into chlorophyll-a during the chlorophyll cycle (Figure 2). The chlorophyllase enzyme catalyzes chlorophyll-a into chlorophyllide-a or pheophytin and subsequently into pheophorbide-a. Pheophorbide-a monooxygenase catalyzes the pheophorbide-a and is converted to red chlorophyll catabolites ensuing fluorescent and non-fluorescent chlorophyll catabolites [77,78]. A high temperature of 42 °C declines the enzymes efficiency viz., 5-aminolevulinate dehydrogenase (45%), mg-protoporphyrin IX methyltransferase (65%), protochlorophyllide oxidoreductase (70%), and increases chlorophyllase (46%) in wheat [79].

Chlorophyll deficiency reduces the absorbance of light energy and transfer to the reaction centers (RCs) of PS-II and PS-I at high temperature in wheat [80,81]. Carotenoids dissipate the excess light and protect the reaction centers against stress conditions [82]. Carotenoids viz., xanthophylls, and isoprene maintain the thylakoid membrane from leakage [83]. However, thylakoid components are sensitive at a temperature above 40 °C and inhibit the carotenoids biosynthesis pathways in the chloroplast [46,84], which interrupt the photosynthesis stability and ultimately reduce the grain yield in wheat [25].

### 2.5. Protein Quality

The protein content, protein quality, and glutenin/gliadin determine the backing quality of bakery products [85,86]. High temperature enhances the total protein content but reduces the end use of protein quality [87,88], which is more or less dependent on the grain protein concentration [89]. Protein fractions (albumin, globulin, gliadin, and glutenin) are important components for the end use quality of wheat grain [90]. High temperature at the grain filling duration decreases the albumin and globulin content [91], whereas it increases the gliadin content at the expense of glutenin content in wheat [92]. Furthermore, high temperature increases the protein content but reduces the production of glutenin, sedimentation index [71], and essential amino acids such as lysine, methionine, and tryptophan content, which determines the viscoelastic properties of wheat loaf [45].

### 2.6. Physiological Process

Heat stress inhibits the photosynthesis, damaging photosynthetic apparatus, and synthesis of ROS (reactive oxygen species) as discussed below.

#### 2.6.1. Photosynthesis Response to High Temperature

A high temperature of 35/25 °C (day/night) at the grain filling duration inhibits the leaf photosynthesis up to 50% in wheat (Figure 3 and Figure 4). The net photosynthesis during the wheat crop cycle is essential in controlling the crop biomass and grain yield under a high temperature. The optimum temperature for net photosynthesis is 20–30 °C, but a high temperature above 32 °C declines the photosynthetic rate rapidly in wheat [46]. The photosynthesis in wheat leaves is more sensitive than those, which are associated with the synthesis and mobilization of stem reserves into developing grains during grain filling. Photosynthesis is associated with the activity of photosynthetic apparatus, Rubisco (Ribulose bisphosphate carboxylase/oxygenase) enzyme, and various green organs of the plants such as chlorophyll content and carotenoids [76,93].

#### 2.6.2. Photosynthetic Apparatus

High temperature disturbs the photosystem-II (PS-II) and photosystem-I (PS-I) mediated electron transport chain (ETC). A high temperature of 35–40 °C at the grain filling phase directly damages the photosynthetic apparatus including the PS-II and PS-I mediated electron transport chain [46]. PS-II is a complex subunit of chlorophylls and proteins and is more sensitive than PS-I [73,95]. It harvests the light energy to oxidize the water molecule and transfer electrons to plastoquinone (PQ) ensuing the cytochrome b6f complex, but a high temperature declines the efficiency of PQ and Cytochrome b6f [96].

The light harvesting complex-II (LHC-II) is an assortment of proteins associated with the PS-II core complex. It harvests the sunlight energy and transfers it to the PS-II core complex to form multi-complex proteins [97]. High basal florescence separates the LHC-II from the PS-II core complex and alters the energy distribution to PS-I [98]. A high temperature of 32–38 °C also synthesizes the zeaxanthin, which destabilizes the thylakoid membrane composition and photosynthetic apparatus [48].

#### 2.6.3. Rubisco Activity

Rubisco is an essential light activated enzyme, which possesses the binding sites for CO_2_ and Rubisco activase for the regulation of the Calvin cycle, but its efficiency gradually declines under a high temperature of 25–40 °C in wheat [99]. Sugar phosphate inhibitors viz., XuBP (D-xylulose-1, 5-bisphosphate), RuBP (Ribulose-1, 5-bisphosphate), CA1P (2-Carboxy-D-arabinitol 1-phosphate), and CTBP (2-Carboxytetritol-1, 4-bisphosphate) impaired with the active site, which modulate the Rubisco activity for photosynthesis [100,101]. Rubisco activase removes these inhibitors from the active site and facilitates the carboxylation reaction modulated by the Rubisco enzyme [102]. It also protects the nascent proteins from aggregation but it is heat labile. Therefore, a high temperature of >32 °C alters the composition for the accessibility of carbamylation [103,104].

High temperature declines the solubility of CO_2_ and enhances the O_2_ level from the compensation point due to the reduction in evapotranspiration [105,106,107] and specificity of the Rubisco enzyme activity, which is poor in discriminating O_2_ and CO_2_ [108,109] (Figure 5). These factors stimulate the photorespiration and consume ATPs, release the fixed CO_2_, and produce the photorespiratory metabolite (glyoxylate), which consume NADH_2_ [110,111] and ultimately reduce the yield up to 20% in wheat [112].

#### 2.6.4. Reactive Oxygen Species

Reactive oxygen species (ROS) are synthesized during the malfunction of PS-II and the Calvin cycle of photosynthesis [113] causing lipid per-oxidation and cell membrane damage in wheat [114,115]. ROS such as super oxides (O^-2^), hydroxyl radical (OH^-^), and hydrogen peroxide (H_2_O_2_) commonly synthesize at high temperatures. The manganese superoxide dismutase (Mn-SOD) catalysis in mitochondria produces hydrogen peroxides, whereas the auto-oxidation of ubisemiquinone complex-I and complex-III generates super oxides radicals ensuing the oxidative stress in the cell, as well as DNA damage, protein modification, and membrane instability [48,116].

Super oxides synthesize by the reduction of one electron, whereas further electrons reduction generates peroxide, which is neutralized by two protons of hydrogen atom and form H_2_O_2_, as shown in Figure 6. Hydrogen peroxide is produced by incomplete water molecules oxidation, which is reduced by the manganese to form the hydroxyl radical [117]. The hydrogen peroxide concentration gradually increases from vegetative to milky dough stage at a high temperature and negatively influences the photosynthesis [118].

## 3. Tolerance Mechanism against High Temperature

The plant’s tolerance to high temperature facilitates adaptation in adverse conditions through maintaining their physiology and ameliorate grain yield.

### 3.1. Phytohormones and Bioregulators

Phytohormones inevitably associated with the antioxidant enzymes activity and growth regulation under heat stress conditions [119]. Phytohormones viz., proline, glycine betaine, salicylic acid, abscisic acid, and ethylene maintain the physiology at a high temperature through soluble salts accumulation in the cell and reducing H_2_O_2_ production in wheat, as displayed in Figure 7.

A high temperature of 30–35 °C discolorizes the chlorophyll, beta-carotene, and damages the photochemical activity. Glycine betaine accumulates in the chloroplast of leaves and stabilizes PS-II, reaction centers in the thylakoid membrane [120,121], Rubisco enzyme, and inhibits the ROS production [122]. It adjusts the osmotic pressure, ameliorate antioxidant enzymes activity, and photosynthesis under high temperature in wheat [123]. Salicylic acid acts as a phenolic hormone in plants and is responsible for osmoregulation, scavenges ROS, and maintains the photosynthesis in wheat [124]. It also triggers the osmolytes synthesis viz., glycine betaine, proline, and sugars under heat stress conditions [125,126,127]. 

Proline accumulation is determined by the proline dehydrogenase activity and Δ1-pyrroline-5-carboxylate synthetase/reductase (P5CS) [128]. High temperature increases the *P5CS* and decreases proline dehydrogenase in tolerant wheat seedlings. Proline dehydrogenase catalyzes the proline degeneration in mitochondria. However, glutamate acts as a precursor in the presence of *P5CS1* for the proline synthesis and accumulates in plant under heat stress conditions [129,130]. The proline content is directly linked to a high temperature of 35–40 °C and ameliorates the defense mechanism in wheat seedlings [131]. A high temperature of 35 °C than 25 °C accumulates a higher proline content (up to 200%) and improves the photosynthetic efficiency and yield [132].

Bioregulators upregulate the antioxidant defense mechanism and maintains the PS-II under high temperature. Foliar application during the grain filling period and seed priming with a 6.6 mM solution of thiourea intensifies the antioxidant enzyme activity, chlorophyll content, total soluble protein, amino acid, and grain weight in wheat [133]. Foliar application of 50 ppm dithiothreitol also ameliorates the adverse effect of high temperature in wheat [134].

### 3.2. Stay Green

Stay green represents the chlorophyll retention and longevity of photosynthetic apparatus for the adaptation of wheat under high temperature [135,136,137]. Stay green associated with the stabilized photosynthetic apparatus of chloroplast viz., scavenges of ROS, and maintaining the photosynthetic apparatus indicates the slow degeneration of tissues in wheat.

The stay green trait has the potential to protect photosystem-II in the chloroplast and inhibits the ROS synthesis in wheat [138,139]. It maintains the green pigment at a high temperature of >30 °C during the grain filling phase. The short grain filling duration and high canopy temperature are associated with non-stay green genotypes in wheat [140]. Stay green is positively associated with the normal grain filling phase, membrane stability, photosynthesis, stem reserve carbohydrates, and grain development [141,142].

Chlorophyll biosynthesis enzymes determine the senescence in wheat, which influences the assimilation and translocation of photosynthates into grains during grain filling [37,143]. For example, the *SGR* mutant of Arabidopsis and rice exhibit the stay green phenotype due to the suppression of Mg dechelatase enzyme, which is responsible for chlorophyll degradation [144]. *SGR* mutants have also been reported in other species viz., pea, tomato, and pepper [142]. The *NYC* gene suppression also delays the senescence of crops that catalyzed the chlorophyll breakdown for the conversion of chlorophyll-b into chlorophyll-a [145]. The *PPH* genes mutant removes the phytol from phaeophytin in Arabidopsis and results in stay green [146]. Genes *NYC*, *PPH,* and *SGR* have a potential role for stay green in arbidopsis and rice that must be explored in wheat to improve thermotolerance.

### 3.3. Antioxidant Enzymes

Antioxidant enzymes protect the plant from ROS, convert the free radicals of oxygen and hydroxyl into H_2_O_2_ followed by the water molecule. These enzymes scavenge the ROS, balance the production/elimination of ROS from oxidative stress, maintain the growth, development, metabolism, and overall productivity [147]. Antioxidant enzymes viz., POD (peroxidase), SOD (superoxide dismutase), CAT (catalase), and GR (glutathione reductase) usually generate under a high temperature of 35/28 °C day/night in wheat [147,148,149].

The SOD enzyme converts the O^−2^ to H_2_O_2_, which is a less toxic form than the free radicals [150,151]. CAT and POD convert H_2_O_2_ into H_2_O, but the CAT activity is higher than other antioxidant enzymes in wheat [152,153]. CAT reduces several millions of H_2_O_2_ molecules into H_2_O and oxygen per minute [154,155]. GR protects the plant from oxidative stress by reducing oxidized glutathione [156,157]. Glutathione peroxidase (GPx) efficiency depends on high γ-glutamyl cysteine synthetase and glutathione synthetase activity for the reduction of H_2_O_2_ into H_2_O [158].

### 3.4. Heat Shock Proteins

Wheat plant produces heat shock proteins (HSPs) at 32–34 °C and provides protection against high temperature [159,160]. High temperature disturbs the membrane proteins in plants but upregulates the translation of heat shock genes, which encodes for HSPs [132,161,162]. These HSPs protect the cell from adverse effects of heat stress by maintaining photosynthesis, upregulation of other proteins, and cell metabolism [163]. There are different families of ATP dependent HSPs viz., HSP60, HSP70, HSP90, and HSP100 except small HSPs based on molecular weight.

The small HSP (smHSP) in wheat genome assembles with other homo-oligomers and facilitates binding in ATP independent manners. It assembles with HSP90 to prevent unfolding and refolding of proteins under high temperature [159,160]. HSP60 expresses constitutively in chloroplast and mitochondria [164,165]. The Rubisco large subunit binding protein (chHSP60) is a cofactor of HSP60, which regulates the Rubisco enzyme folding at high temperature [166].

HSP70 is a highly conserved protein, which recognizes only a short sequence of the polypeptide chain, temporal and inhibits aggregation of non-native protein at high temperature [167]. HSP110 is a sub family of HSP70 and inhibits the aggregation with a greater capability than HSP70 [168]. HSP90 regulates transcription, cellular signaling, and managing protein folding through assembling molecular proteins including HSP40 and HSP70 [118,168,169], whereas HSP100 interacts with different smHSPs and HSP70 to prevent the aggregation of protein [170].

## 4. Tolerance Strategies against High Temperature

Strategies against heat stress viz., crop management, conventional, non-conventional, and molecular approaches ameliorate the thermotolerance in wheat. These strategies are further elaborated below.

### 4.1. Crop Management

Agronomic practices including seed priming, organic mulches, inorganic fertilizers, and timely sowing with recommended management practices mitigate the heat stress in wheat. Wheat seed priming in the aerated solution of CaCl_2_ (1.2%) for 12 h improves the germination, growth, leaf area index, chlorophyll content, assimilation rate, and grain yield [171,172,173]. Mulching with rice husk conserves water, improves water use efficiency, maintains the water status in soil, and slows down the release of nitrogen for plant uptake [174,175].

The application of inorganic fertilizers viz., nitrogen, and potassium maintain the chlorophyll content, osmoregulation, cytokinin biosynthesis, protein stability, redox homeostasis, and photosynthesis at high temperature [25,176]. Zinc improves the superoxide dismutase activity, membrane integrity, chlorophyll content, chlorophyll florescence, and kernel growth at high temperature [27,177]. The silicon application at 10 mg/kg of soil at heading improves the osmotic potential (26%), photosynthetic rate (21%), catalase activity (38%), superoxide dismutase activity (35%), stomatal conductance (20%), and transpiration rate (32%) in wheat under high temperature [178,179].

Sowing time is a counteract strategy against high temperature. Delayed planting compels the plants to complete their growing degree days earlier, but they have to face high temperature during the anthesis and grain filling phase [53,180]. Wheat planted in normal sowing dates utilizes a longer duration to capture the available reserves/carbohydrates and improve the grain development [70,181,182].

### 4.2. Conventional Approaches for Thermotolerance

Thermotolerance is an inherited component stabilizing economic yield against heat stress. Tolerance to high temperature is characterized as the least effect on growth, development, and productivity. Screening of wheat genotypes is difficult in a spatial environment under natural heat stress conditions. This is due to the consistent selection criteria that have not been developed to screen diverse germplasm. The selection criteria based on traits directly associated with the grain yield facilitates better improvement in the genetic material for thermotolerance (Table 2).

Breeding has made considerable advances in the genetic basis, diversity, and development of thermotolerant varieties. However, utilization and explorations of novel genetic diversity facilitates the genetic improvement for thermotolerance in the breeding program. However, the genetic gain is limited due to the narrow genetic basis [183,184]. Therefore, utilizing landraces and wild relatives increases the genetic variation in wheat for developing thermotolerance. Breeding for thermotolerance utilizing land races and wild relatives viz., *Aegilops speltoides*, *Aegilops tauschii*, *Triticum turgidum,* and *Triticum durum* have the ability to maintain chlorophyll content, canopy temperature depression, membrane stability, and photosynthesis under stress conditions [74,185,186,187].

**Table 2 plants-10-00043-t002:** Major desirable selection criteria for the screening heat tolerance in wheat.

Traits	References
Cell membrane stability	[50,51,188,189]
Proline content	[131,190,191,192]
Heat susceptibility index for grain yield	[25,193,194,195,196]
Chlorophyll content	[76,131,188,189,197,198]
Photosynthesis	[48,106,107,117,199]
Stay green	[70,137,140,142,143,200]
Grain filling duration	[70,181,201]
Grains formation	[59,67,202,203,204]
Grain development	[67,71,203,204,205]
Early heading	[46,64,204,206,207]
Canopy temperature depression	[30,140,201,208,209,210,211,212]

### 4.3. Non-Conventional Approaches

Plants development utilizing genetic engineering or the indirect selection of traits through molecular markers or omics technology facilitates the improvement against heat stress in wheat.

#### 4.3.1. Biotechnological Approach and Heat Shock Factors

Genetic engineering is the development of cultivar through incorporating the individual gene [213]. Advances in biotechnology enable the faster genetic gain than conventional breeding methods. Several genes encoding heat shock factors have been identified in wheat, but novel genes identification for thermotolerance remains a challenge (Table 3). The identification of novel genes and their altered expression under high temperature in wheat crop provides the molecular basis for improving thermotolerance.

##### Quantitative Trait Loci (QTL)

Heat tolerance is under polygenic control and the QTL analysis enlightens the genetic basis of thermotolerance in wheat. It facilitates the indirect selection of traits rather than the selection based on phenotypic traits. Many QTLs have been identified for physio-morphic traits in wheat, but few were identified against heat stress (Table 4), which facilitates in gene pyramiding and marker assisted selection in wheat breeding programs for developing thermotolerance. 

#### 4.3.2. Omics Technology

Omics techniques facilitate the development of thermotolerance in wheat through the identification of transcriptional, translational, and post translational mechanisms (Figure 8). Transcriptomics represent the alteration in transcriptome factors under different environmental conditions through the DNA microarray technology [235,236]. It has already been used to study the glumes [237], grain development [238], and quality traits [239] for the identification of candidate gene expression under heat stress conditions. MicroRNAs (miRNAs) are non-coding small RNA that serve as the regulation of post-transcriptional gene expression in plants. Micromics assist in the candidate miRNA identification and their role in transcriptome homeostasis, developmental, and cellular tolerance of plants under high temperature [197].

Proteomics is the analysis of candidate proteins, the expression when they translated from mRNA to functional proteins, and a further characterization of their role in the heat tolerance mechanism [240]. Proteomic analysis revealed heat shock proteins, protein synthesis, detoxification, photosynthesis, and protein quality under heat stress conditions [115,241,242,243,244,245]. Hence, the omics technology provides us with a novel opportunity for the identification of genes, their expression, and pathways linked to these genes. However, the further genetic network and their component identification will be a challenge to adapt plants in a high temperature environment.

## 5. Conclusions and Future Prospects

Temperature is gradually increasing and affecting crop productivity. The impact of high temperature on wheat crop has been extensively studied, but understanding the mechanism of thermotolerance remains elusive. High temperature disrupts membrane stability, declines grain filling duration, grain formation, and starch accumulation into grains. Inhibition in the physiological process has been observed due to the high temperature stress. It disturbs the photosynthetic apparatus and generates the reactive oxygen species leading towards oxidative stress. The strategy against high temperature requires systematically understanding the physiological, metabolic, and development process associated with thermotolerance. The tolerance mechanism including more accumulation of proline, glycine betaine, antioxidant enzyme activity, heat shock protein, and stay green could be a useful indicator for thermotolerance.

Crop management stabilizes the physiological process and metabolic pathways through mulches, extra irrigation, inorganic fertilizers, early sowing, exogenous application of micronutrient, osmoprotectants, and bioregulators. Integrating crop management practices with molecular genetics tools can ameliorate the adverse effects of high temperature, but need to further explore the strategies associated with high yield under heat stress [246,247,248,249,250,251,252,253,254,255,256,257,258]. The tolerance development can be achieved through a selection based on thermotolerant traits from existing germplasm and breeding utilizing land races and their wild relatives. The suitable selection criteria based on thermotolerant traits requires developing germplasm against heat stress. Recently, the canopy temperature depression at the reproductive stage, grain filling duration, heat susceptibility index for grain yield, and stay green have been established for screening germplasm against heat stress conditions. Stay green with other useful traits provide the solution of the burning problem due to the high temperature in wheat. Therefore, the contribution of the synthesis of chlorophyll turnover equation in photosynthesizing leaves for the stay green trait expression has a good future against high temperature stress.

The marker assisted breeding programs must be pooled with the transgenic approach for thermotolerance QTLs and genes. Understanding the QTLs and omics techniques pave the way to develop thermotolerance in wheat, but a further understanding of the genes network and their regulation of expression related to high temperature would be a challenge. There is a need to understand the molecular and biochemical basis of thermotolerance from the upcoming changing climate for crop improvement. Functional genomics also proved to be supportive against high temperature, but the alteration in transcriptomes and proteomes needs to be further investigated against high temperature. Noteworthy, molecular and genetic approaches facilitate crop adaptability coupled with the economic yield under high temperature, but the expression of yield potential requires the estimation of yield at the crop level. Therefore, the application of incorporating a future scenario into crop models provides model-based recommendations to improve thermotolerance in wheat.

## Figures and Tables

**Figure 1 plants-10-00043-f001:**
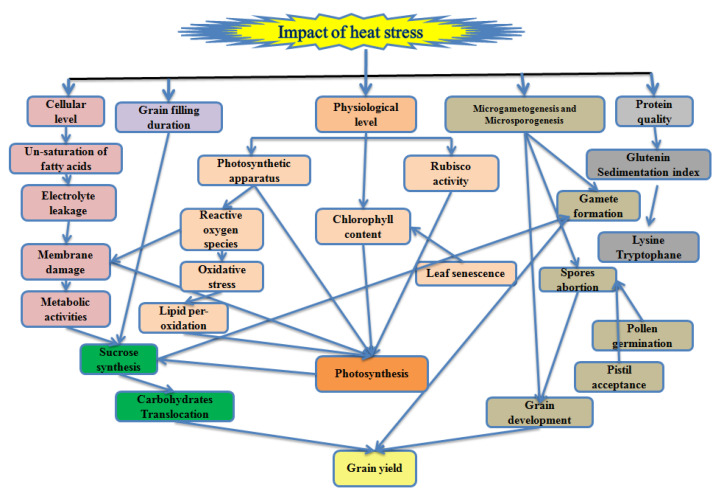
Schematic illustration of the high temperature impact on wheat associated with the grain yield.

**Figure 2 plants-10-00043-f002:**
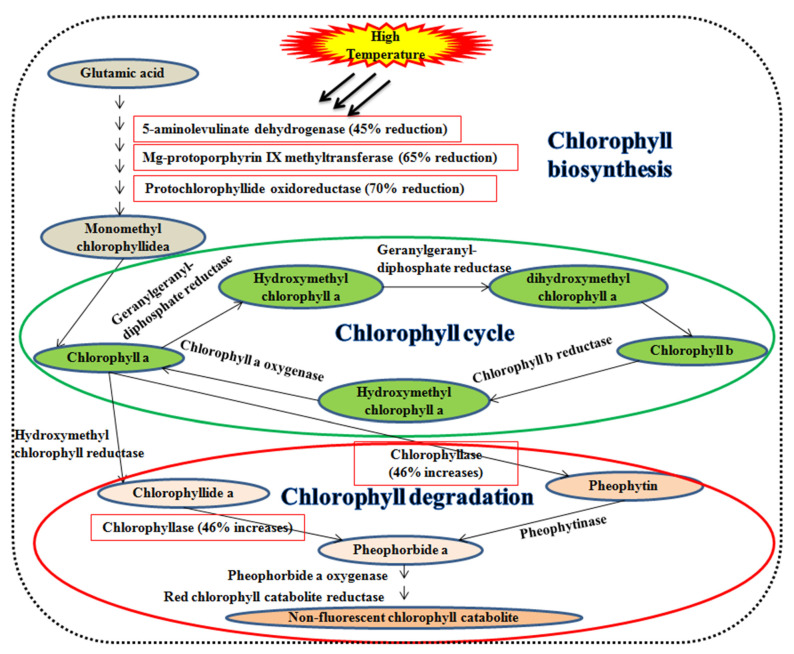
Impact on the high temperature of leaf senescence. Enzymes associated with chlorophyll synthesis viz., 5-aminolevulinate dehydrogenase, mg-protoporphyrin IX methyltransferase, and protochlorophyllide oxidoreductase, whereas chlorophyllase is responsible for chlorophyll degradation.

**Figure 3 plants-10-00043-f003:**
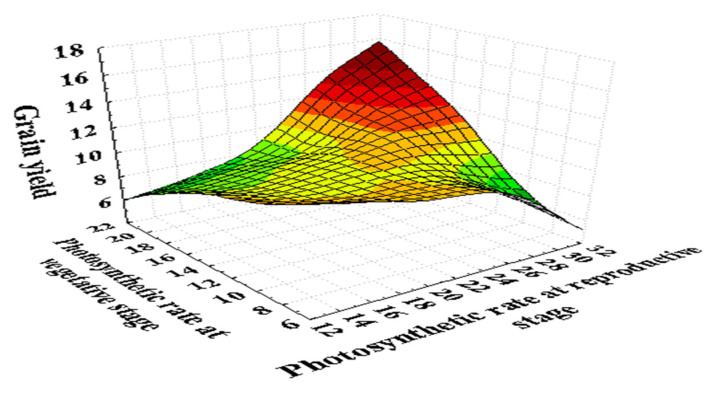
Photosynthetic (µmol m^−2^ s^−1^) response at the seedling and reproductive stage of 180 wheat genotypes with the grain yield per plant (g). Photosynthetic rate was recorded on a clear day between 10:00 a.m. to 12:00 p.m. with the help of infrared gas analyzer (IRGA ADC, LCA-4, Hoddesdon, UK). Data collected under normal and heat stress conditions at the vegetative (Zadoks scale 39) and reproductive stages (Zadoks scale 69) [94]. It represented that the photosynthesis is directly associated with the grain yield at both stages. As the photosynthetic rate decreases, it reduces the grain yield in wheat.

**Figure 4 plants-10-00043-f004:**
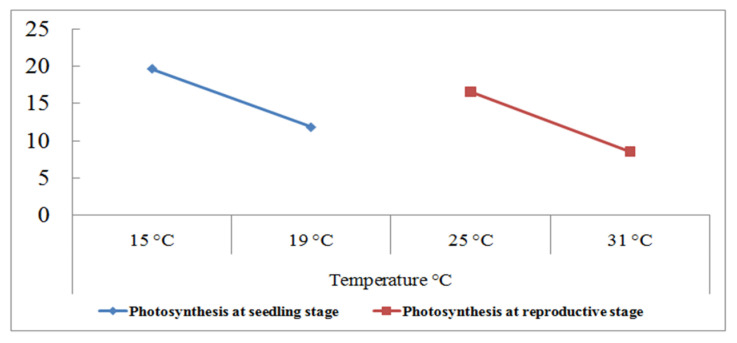
Photosynthetic (µmol m^−2^ s^−1^) response of 180 wheat genotypes with the grain yield per plant (g) at the seedling and reproductive stages. Data collected under normal and heat stress (4–5 °C above normal) conditions [94].

**Figure 5 plants-10-00043-f005:**
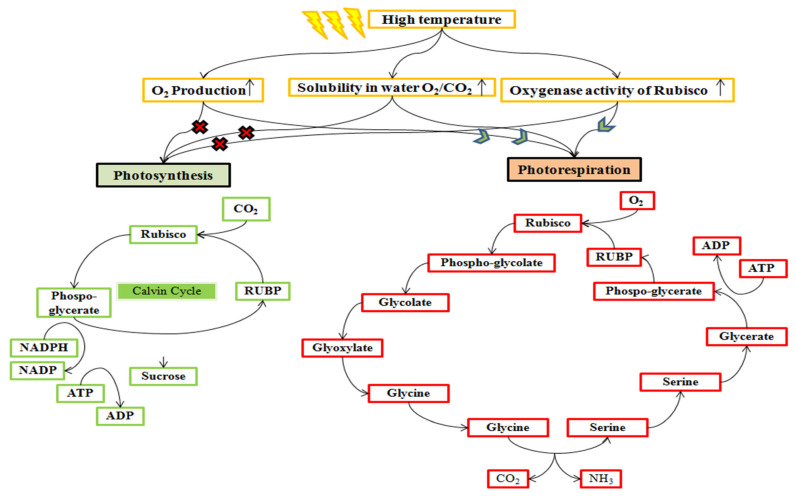
Rubisco enzyme activity pathway alteration at a high temperature. Rubisco has a characteristic of both oxygenation and carboxylation activities. High temperature increases the synthesis of oxygen through photosynthesis, which enhanced the solubility of oxygen than carbon dioxide. Therefore, it promotes the oxygenase activity of Rubisco and stimulates photorespiration, which compartmentalized in chloroplast, peroxisomes, and mitochondria.

**Figure 6 plants-10-00043-f006:**
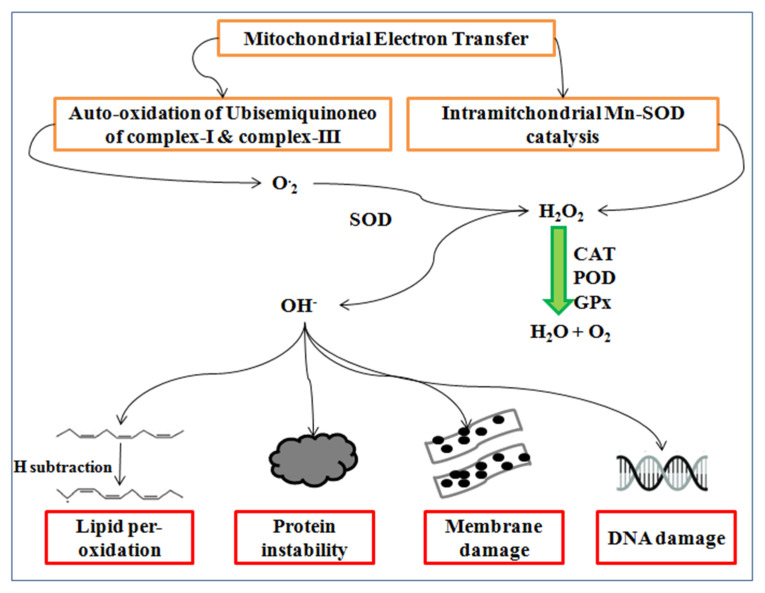
Synthesis of the reactive oxygen species and their consequences.

**Figure 7 plants-10-00043-f007:**
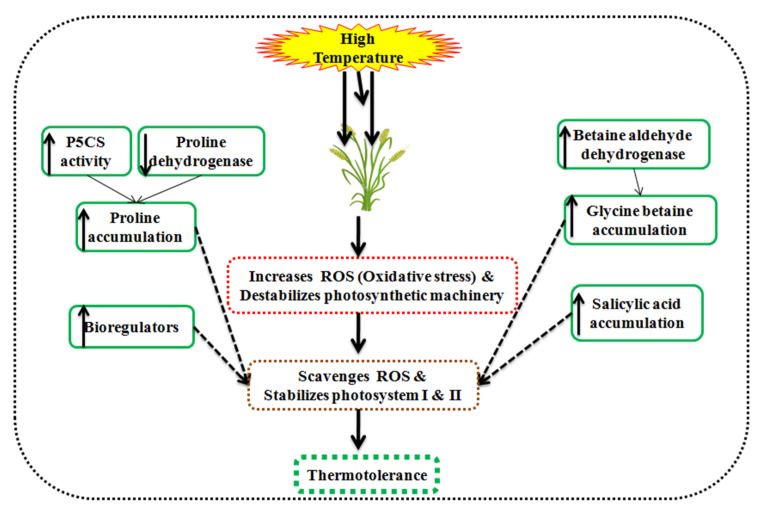
Schematic illustration of osmolytes associated with thermotolerance in wheat.

**Figure 8 plants-10-00043-f008:**
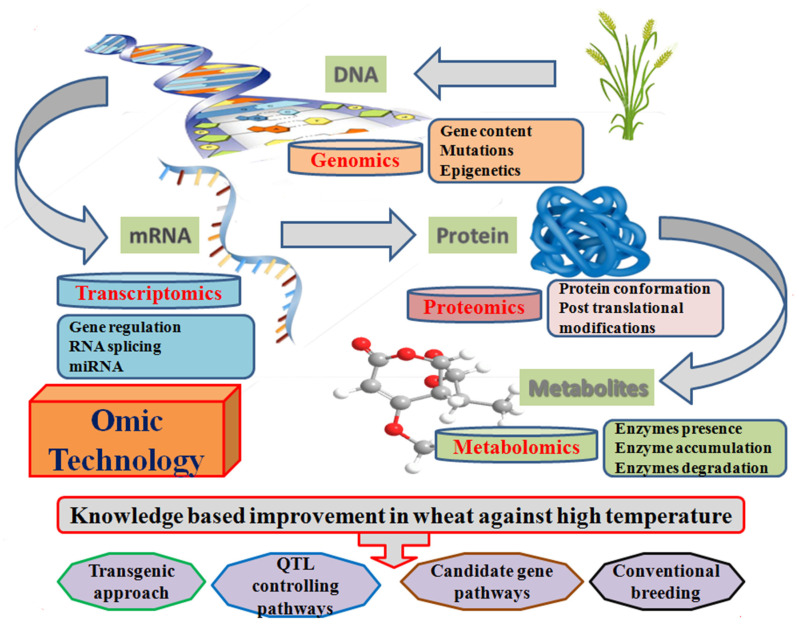
Schematic diagram representing the omics techniques associated with thermotolerance development in wheat at the molecular genetics level.

**Table 1 plants-10-00043-t001:** Optimal temperature requirements of wheat at different growth stages.

Stages	Optimum Temperature (°C)	Minimum Temperature (°C)	Maximum Temperature (°C)
Root growth	17.2 ± 0.87	3.5 ± 0.73	24 ± 1.21
Shoot growth	18.5 ± 1.90	4.5 ± 0.76	20.1 ± 0.64
Leaf initiation	20.5 ± 1.25	1.5 ± 0.52	23.5 ± 0.95
Terminal Spikelet	16 ± 2.3	2.5 ± 0.49	20 ± 1.6
Anthesis	23 ± 1.75	10 ± 1.12	26 ± 1.01
Grain Filling Duration	26 ± 1.53	13 ± 1.45	30 ± 2.13

[28,30,31,32,33,34,35,36,37,38,39,40,41,42,43,44,45,46,47].

**Table 3 plants-10-00043-t003:** List of genes encoding transcription factors related to thermotolerance.

TFs/Genes	Source	Function	Reference
*Hsf6A*/wheat	HVA1s promoter of barley	Regulation of heat shock protein genes and improve thermotolerance	[214]
*EF-Tu*	Ubiquitin 1 promoter of maize	Overexpression reduces the thermal aggregation of leaf proteins, photosynthetic membrane, and increases CO_2_ fixation	[215]
*HvSUT1*	Hordein B1 promoter of barley	Increase sucrose transport into grains	[216]
*TaFER-5B*	Ubiquitin 1 of maize	Reduces oxidative stress by scavenging ROS and improves leaf iron content	[217]
*TaGASR1*	Wheat variety TM107	Reduces ROS and hormonal signal transduction pathway	[218]
*TaHsfC2a*	Monocot-specific HsfC2 subclass	Thermotolerance development via the ABA-mediated regulatory pathway	[219]
*TaHSP23.9*	Chinese wheat based on proteomic analysis	Upregulation under heat stress facilitates in seed development during the grain filling phase	[220]
*TaFBA1*	F-box gene from wheat	Upregulation improves photosynthesis and the antioxidant enzyme activity	[221]
*TaHsfA2-1*	Wheat	Overexpression of heat shock proteins and chlorophyll content	[222]
*SGR*	Arbidopsis and rice	Binding of light harvesting complex during photosystem-II	[142,144]
*NYC*	Arbidopsis and rice	Responsible for the activity of chlorophyll reductase to convert chl-b into chl-a	[142,145]
*PPH*	Arbidopsis and rice	Responsible for the activity pheophytinase for dephytylation to phaeophytin	[142,146]

**Table 4 plants-10-00043-t004:** Major quantitative trait loci (QTL) identified for traits against heat stress.

Traits	Chromosome	References
Chlorophyll content	*2A, 3A, 6A, 7A, 2B, 5B, 2D*	[223,224]
Chlorophyll florescence	*1A, 2A, 3A, 3B, 2D, 1D*	[224,225]
Plasma membrane damage	*7A, 2B*	[223]
Thylakoid membrane damage	*6A, 7A, 1D*	[223]
Canopy temperature	*7A, 1B, 2B, 3B*	[226]
Grain weight	*1A, 2A, 4A, 1B, 2B, 3B, 4B, 6B, 6D*	[226,227,228]
Grains formation	*1A, 4A, 2B, 3B, 5B*	[228,229]
Chlorophyll florescence	*1A, 4A, 1B, 2B, 7D*	[230]
Senescence	*2A, 3A, 6A, 7A, 3B, 6B*	[231,232]
Stay green	*1A, 3B, 7D*	[233,234]

## Data Availability

The data presented in this study are available in this study as well as in the PhD thesis of Adeel Khan i.e., first author.

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
