# Peer review of "Rising Atmospheric Temperature Impact on Wheat and Thermotolerance Strategies"

_plants, 2020, doi:10.3390/plants10010043_

Round 1

Reviewer 1 Report

The impact of high temperature on wheat growth and development is an important topic for agriculture, which is worthy of reviewing. However, the present manuscript is not suitable for publication because of various problems. 

(1) the mistakes in text should be dealt with. e.g. "Global warming potential of carbon dioxide is 1, methane 28-36 and nitrous oxide 265-298 over hundred years." (line 47-48); "Phytoharmones" is not correct (line 211, 213 etc.); please check all the text.

(2) There are too many unnecessary schemes, e.g. Fig. 4, Fig. 6, Fig. 8 and Fig. 9. These figures are too simple and already well-known.

(3) The writting language should be concise, which would make the manuscript more readable

Author Response

REPLY TO REVIEWER#1

Comments

Reply to Reviewer Comments

Page and Line No.

Reviewer 1

The main problem with this review is that it only summarizes the previous research without any suggestions and orientations for future studies. There is a lack of discussion, author view and suggestions. The paper did not provide the prospect and challenge. There does not seem to be a novel angle that might help drive future research in a certain direction. The submission is not clear in how it can significantly advance our mechanistic understanding of the topic nor any aims or objectives are provided. I would like to suggest the authors to provide an in-depth review and focus on the importance of the topic and identify the knowledge gaps in this area of research.

We have revised the manuscript in the lights off suggested valuable comments and added the future prospect for improving thermotolerance in wheat. In this manuscript we focus on the impact of high temperature on wheat crop and suitable strategies to cope with heat stress based on previous experiments. On the basis of these we explore the new areas to be focused in future to upcoming scenario of high temperature.

Page No. 15

Line No. 387-419

Specific Comments (Check spellings)

indispensible role; check spelling for indispensable

Word indispensible is replace by indispensable

Page No. 4

Line No. 116

carotenoids biosynthatase pathway?

Word biosynthatase is replace by biosynthesis

Page No. 4

Line No. 131

5-aminolevulinate dehygrogenase?

Word dehygrogenase is replace by dehydrogenase

Page No. 5

Line No. 135

Mg-protoporphyrin IX methyltransferease?

Word methyltransferease is replace by methyltransferase

Page No. 5

Line No. 135

Cyto-b6f write in full; cytochrome-b6f.

Word Cyto-b6f is replace by cytochrome-b6f

Page No. 7

Line No. 180

Phytoharmones, check spelling?

Word Phytoharmones is replace by Phytohormones

Page No. 8

Line No. 225-227

Cholorplast?

Word Cholorplast is replace by Chloroplast?

Page No. 9

Line No. 231

gluthatione synthetase? Check spelling.

Word gluthatione is replace by glutathione

Page No. 11

Line No. 287

….. their growing degreed days….?

Word degreed is replace by degree

Page No. 12

Line No. 328

physoio-morphic traits?

Word physoio-morphic is replace by physio-morphic

Page No. 13

Line No. 361

glycine btaine?

Word btaine is replace by betaine

Page No. 15

Line No. 394

Thermotoleracne?

Word Thermotoleracne is replace by Thermotolerance

Page No. 15

Line No. 411

Reviewer 2 Report

In this review Khan et al. described the impacts of rising atmospheric temperature on wheat and thermotolerance strategies. The topic is of fundamental importance however, the review is incomplete in many ways. The main problem with this review is that it only summarizes the previous research without any suggestions and orientations for future studies. There is a lack of discussion, author view and suggestions. The paper did not provide the prospect and challenge.  There does not seem to be a novel angle that might help drive future research in a certain direction. The submission is not clear in how it can significantly advance our mechanistic understanding of the topic nor any aims or objectives are provided. I would like to suggest the authors to provide an in-depth review and focus on the importance of the topic and identify the knowledge gaps in this area of research. Beside this, check and correct the following mistakes:

indispensible role; check spelling for indispensable.

carotenoids biosynthatase pathway?

5-aminolevulinate dehygrogenase?

Mg-protoporphyrin IX methyltransferease?

Cyto-b6f write in full; cytochrome-b6f.

Phytoharmones, check spelling?

Cholorplast?

gluthatione synthetase? Check spelling.

….. their growing degreed days….?

physoio-morphic traits?

glycine btaine?

Thermotoleracne?

Author Response

REPLY TO REVIEWR#2

Reviewer 2

Comments

Reply to Reviewer Comments

Page and Line No.

L 55: You omitted the full name for NOAA (National Oceanic and Atmospheric Administration). Please provide it, with abbreviations defined in parentheses, as needed according to instructions for authors.

We have provided the full name of NOAA

Page No. 2

Line No. 57-58

L 64: I don’t see the need for table 1. The subject of this review paper is very extensive, and the rest of the text is focused solely on wheat, so this table kind of seems redundant.

WE removed the Table 1 from the manuscript according to reviewer suggestion because this manuscript focuses on the wheat crop.

Page No. 2

Line No. 67

L 82-89: In chapter 2.1. (Cellular metabolism) you write about how heat causes electrolyte leakage, but what is the end result? This paper summarises the basic knowledge on the impact of rising temperatures on wheat, so such cause and effect relations should be emphasised.

We have added the effect of this electrolyte leakage associated with grain yield in wheat.

Page No. 4

Line No. 92-94

L 146-150: Chapter 2.6.1. (Photosynthesis response to high temperature) is a bit short on information. It would be good if you could maybe provide more concrete references on the overall effect of heat stress on photosynthesis, before getting into explaining different aspects of the process such as the functioning of the photosynthetic apparatus, the role of Rubisco and ROS molecules.

We have provided more references regarding the impact of high temperature on photosynthesis

Page No. 6

Line No. 153-158

L 152-155: Is figure 3 showing the results of some earlier published research? If so, please provide the reference; if not, please provide more information about the research.

We have collected the data from Author Adeel Khan (Ph.D Thesis) and added this information in the caption of figure3.

Page No. 6

Line No. 163-167

L 156-159: The same as for figure 3.

We have provided the reference where we collected the data of photosynthetic rate and grain yield

Page No. 6

Line No. 171-173

L 197: You omitted the full name for Mn-SOD. Please provide it, as you did for other enzymes.

We have added the name of Mn-SOD.

Page No. 8

Line No. 211

L 236-245: In chapter 3.2. (Stay green) you write about the staygreen trait in wheat but you don’t mention its genetic background. Please elaborate on this.

We have provided the genetic background of stay green with references

Page No. 9

Line No. 261-268

L 248-252: Figure 8 is redundant as it gives no other information but the difference in colour between stay-green genotype and genotype with the onset of leaf senescence. If you want to keep it, please consider omitting the graphics.

We have removed this figure from the manuscript due low graphics

Page No. 10

Line No. 271-275

L 271: You omitted the full name for HSP. Please provide it in full, with the abbreviation in brackets.

We have provided the full name of HSPs as heat shock proteins

Page No. 11

Line No. 292

L 361-362: Figure 10 contains two graphics that are almost the same, please remove one.

There was a repetition of figure 10 mistakenly. So we have removed it from the manuscript

Page No. 14

Line No. 381

L 382-921: Please revise the references according to instructions for authors and be sure to check all the Latin names, as some of them are not in italic.

We revised the manuscript according to your suggestion and made these changes in the manuscript

L 387-389: Please use the correct way to cite references 3 and 4.

We have corrected it from the manuscript

Page No. 15-16

Line No. 431-434

L 397: Please use the correct citation for reference 8.

We added the correct citation of this reference

Page No. 16

Line No. 442-443

Reviewer 3 Report

Dear Authors,

You provided a concise outlook and literature review on a very important, extensive, and complex topic of wheat thermotolerance. It is evident that a lot of work was put into this manuscript, it shows the depth of authors’ knowledge on the subject, and it would be greatly beneficial for someone new to the field. Unfortunately, language shortcomings make it difficult to follow and results in ambiguities. Bluntly put, the information is there, but it is lost in translation. Furthermore, the paper should be adapted according to the instructions for authors, as there are some discrepancies. Some of my other, smaller concerns are:

L 55: You omitted the full name for NOAA (National Oceanic and Atmospheric Administration). Please provide it, with abbreviations defined in parentheses, as needed according to instructions for authors.

L 64: I don’t see the need for table 1. The subject of this review paper is very extensive, and the rest of the text is focused solely on wheat, so this table kind of seems redundant.

L 82-89: In chapter 2.1. (Cellular metabolism) you write about how heat causes electrolyte leakage, but what is the end result? This paper summarises the basic knowledge on the impact of rising temperatures on wheat, so such cause and effect relations should be emphasised.

L 146-150:  Chapter 2.6.1. (Photosynthesis response to high temperature) is a bit short on information. It would be good if you could maybe provide more concrete references on the overall effect of heat stress on photosynthesis, before getting into explaining different aspects of the process such as the functioning of the photosynthetic apparatus, the role of Rubisco and ROS molecules.

L 152-155: Is figure 3 showing the results of some earlier published research? If so, please provide the reference; if not, please provide more information about the research.

L 156-159: The same as for figure 3.

L 197: You omitted the full name for Mn-SOD. Please provide it, as you did for other enzymes.

L 236-245: In chapter 3.2. (Stay green) you write about the stay-green trait in wheat but you don’t mention its genetic background. Please elaborate on this.

L 248-252: Figure 8 is redundant as it gives no other information but the difference in colour between stay-green genotype and genotype with the onset of leaf senescence. If you want to keep it, please consider omitting the graphics.

L 271: You omitted the full name for HSP. Please provide it in full, with the abbreviation in brackets.

L 361-362: Figure 10 contains two graphics that are almost the same, please remove one.

L 382-921: Please revise the references according to instructions for authors and be sure to check all the Latin names, as some of them are not in italic.

L 387-389: Please use the correct way to cite references 3 and 4.

L 397: Please use the correct citation for reference 8.

I hope these comments will be useful.

Best wishes in your future work and kind regards,

Reviewer

Author Response

REPLY TO REVIEWER#3

Reviewer 3

Comments

Reply to Reviewer Comments

Page and Line No.

(1) the mistakes in text should be dealt with. e.g. "Global warming potential of carbon dioxide is 1, methane 28-36 and nitrous oxide 265-298 over hundred years." (line 47-48); "Phytoharmones" is not correct (line 211, 213 etc.); please check all the text.

We have provided the more clear explanation about global warming potential.

Phytoharmones word is replace by phytohormone

Page No. 2 & 8

Line No. 46-49 & 225-227

(2) There are too many unnecessary schemes, e.g. Fig. 4, Fig. 6, Fig. 8 and Fig. 9. These figures are too simple and already wellknown.

Fig 4 explains about impact of increasing temperature on grain yield based on data collected from Authors Ph.D. Thesis that why we added in the manuscript to explain how much photosynthesis associated with grain yield in wheat.

Fig. 6 explains well for the reader about source and mechanism of ROS synthesis and scavenges by antioxidant enzymes.

Fig. 8 and Fig. 9 have been removed from the manuscript as suggested by the valuable reviewer.

(3) The writing language should be concise, which would make the manuscript more readable

Suggestions incorporated as suggested by the valuable reviewer.

Response of wheat to high temperature is a broad topic. To improve thermotolerance in wheat we have to cover all the aspects viz., morphological, physiological, biochemical processes and metabolic pathways. So that we can develop the suitable strategy using transgenic or molecular approach to improve thermotolerance.

Round 2

Reviewer 2 Report

The authors have responded suitably to all my comments raised under review.

Author Response

Author's Reply to the Review Report (Reviewer 2)

Open Review

English language and style

( ) Extensive editing of English language and style required
( ) Moderate English changes required
(x) English language and style are fine/minor spell check required
( ) I don't feel qualified to judge about the English language and style

Reply to Reviewer Comments:

English language and minor spelling as suggested been checked.

Is the work a significant contribution to the field?

Is the work well organized and comprehensively described?

Is the work scientifically sound and not misleading?

Are there appropriate and adequate references to related and previous work?

Is the English used correct and readable?

Comments and Suggestions for Authors

The authors have responded suitably to all my comments raised under review.

Reply to Reviewer Comments:

Thanks for recommending our work.

Reviewer 3 Report

Dear Authors, 

I'm glad to see you accepted my suggestions, however the language, i.e. grammar and style, still need to be corrected. This was one of my major concerns in the first round of reviews as it resulted in many misunderstandings and made the manuscript hard to read, which is still the case. Given the English editing is a part of the submission process, I believe, this will be taken care of in due time. As I had written in the 1st round of reviews, this paper provides a good starting point for those new to the field of researching temperature stress in wheat, as it compiles relevant recent literature, but it doesn’t give much more. This being noted, I will have to leave it on the Editor to decide if the paper is a good enough fit for Plants journal.

Sincerely,

Reviewer

Author Response

Author's Reply to the Review Report (Reviewer 3)

Open Review

English language and style

(x) Extensive editing of English language and style required
( ) Moderate English changes required
( ) English language and style are fine/minor spell check required
( ) I don't feel qualified to judge about the English language and style

Reply to Reviewer Comments:

English language has been improved as suggested by the valuable reviewer after asking English native speaker from Washington State University Pullman USA.

Is the work a significant contribution to the field?

Is the work well organized and comprehensively described?

Is the work scientifically sound and not misleading?

Are there appropriate and adequate references to related and previous work?

Is the English used correct and readable?

Comments and Suggestions for Authors

Dear Authors, 

I'm glad to see you accepted my suggestions, however the language, i.e., grammar and style, still need to be corrected. This was one of my major concerns in the first round of reviews as it resulted in many misunderstandings and made the manuscript hard to read, which is still the case. Given the English editing is a part of the submission process, I believe, this will be taken care of in due time. As I had written in the 1st round of reviews, this paper provides a good starting point for those new to the field of researching temperature stress in wheat, as it compiles relevant recent literature, but it doesn’t give much more. This being noted, I will have to leave it on the Editor to decide if the paper is a good enough fit for Plants journal.

Sincerely,

Reply to Reviewer Comments:

English language has been improved as suggested by the valuable reviewer after asking English native speaker from Washington State University Pullman USA.

We further appreciate comments made by reviewer about work that it provides relevant literature about heat stress. However about reviewer this comment that it does not give much more we will refer reviewer to see section 4 which is about Tolerance strategies against high temperature.
